# Structural, Morphological and Optical Properties of MoS$_2$-Based Materials for Photocatalytic Degradation of Organic Dye

Jadan Resnik Jaleel UC [1], Madhushree R [1], Sunaja Devi K R [1,*], Dephan Pinheiro [1] and Mothi Krishna Mohan [2]

1. Department of Chemistry, CHRIST (Deemed to be University), Bangalore 560029, India
2. Department of Sciences and Humanities, School of Engineering and Technology, CHRIST (Deemed to be University), Kumbalagodu, Mysore Road, Bangalore 560074, India
* Correspondence: sunajadevi.kr@christuniversity.in

**Abstract:** Molybdenum disulfide (MoS$_2$) is a transition metal dichalcogenide (TMDCs) having versatile properties and plays a great role in the photodegradation of organic dyes. MoS$_2$ also finds applications in diverse fields such as catalysis, electronics, and nanomedicine transportation. MoS$_2$ can be prepared by using chemical and physical methods such as hydrothermal, solvothermal, and chemical vapour deposition methods. The preparation method employed can produce subtle but significant changes in the morphology. To increase the efficiency of MoS$_2$, it can be combined with different materials to produce composites that improve the photodegradation efficiency of MoS$_2$. The various methods of preparation, the morphology of MoS$_2$, and photodegradation activity of the MoS$_2$-based nanocomposites are briefly discussed in this review.

**Keywords:** MoS$_2$ nanomaterial; photocatalysis; dye degradation; nanocomposites

## 1. Introduction

The discovery of Buckminster fullerene (C$_{60}$) and carbon nanotubes (CNTs) in the late twentieth century marked a pivotal moment in nanomaterial research. These materials have since become a substitute for carbon-based bulk materials such as graphite and diamond. These materials have already made their presence felt and proved their utility in diverse fields [1]. A significant amount of studies have been done on the carbon nanotube as a versatile material in electronics and numerous other fields [2]. However, the inherent drawbacks in carbon nanotubes led to the discovery of materials like graphene [3], two-dimensional transition metal dichalcogenides (TMDCs), etc. The TMDCs are composed of a transition metal element (M) and chalcogen element (X) like S, Se, Te, etc. [4]. The layers are held together by the weak van der Waals forces. MoS$_2$ is a TMDCs having single-layer/monolayer or nanolayers formed from bulk material via exfoliation. [5]. There are about 40 different types of TMDCs [6], such as WSe$_2$, MoS$_2$, WS$_2$, etc. They are prepared by sandwiching a layer of transition metal (e.g., molybdenum, tungsten, niobium) atoms between two layers of chalcogen (e.g., sulphur, selenium, tellurium) atoms. MoS$_2$ is three atoms thick with a structure comprising of (X-M-X) S-Mo-S [7,8]. There are four different polytypes of MoS$_2$, namely, 1T, 1H, 2H, and 3R. These are classified based on the coordination of Mo atom and stacking orientation of the single layers [6]. MoS$_2$ has found extensive applications in photocatalysis and electrocatalysis [9,10]. MoS$_2$ and MoS$_2$-based nanomaterials have attracted a lot of attention for their high charge carrier potency and superb optical absorption property [11,12]. For commercial photocatalysis applications, a stable, efficient, and low-cost photocatalyst with a light-harvesting spectrum spanning the entire solar spectrum is highly desirable [13]. Photocatalytic materials can transform sunlight into chemical energy and use this to degrade organic pollutants into non-toxic carbon dioxide and tiny inorganic molecules like water without causing secondary pollution [14].

The poor interactions between the layers enable the formation of bulk equivalents of multi-layered materials, which support the stability of the structure. The interesting properties of TMDCs such as tuneable band gap ranging between 0–3 eV, high surface-to-volume ratio, semiconductor property, and the high availability of active sites on the surface have led to a wide range of applications such as gas sensing, optoelectronics, catalysis, energy storage, etc. [15]. The physical properties of these materials are amenable to alteration by inter-lamellar space intercalating or doping [5]. $MoS_2$ has a role in many fields such as catalysis, electronics [16–18], and nanomedicine transportation [19]. $g-C_3N_4$ is another material frequently utilized to alter the characteristics of $MoS_2$ in order to boost its catalytic efficiency [20].

The characteristics of single-layered $MoS_2$ differ significantly from those of their bulk counterparts. The large surface energy in $MoS_2$ makes it prone to aggregation, which can adversely impact its stability and efficiency as a catalyst [21,22]. This can be a serious drawback when $MoS_2$-based catalysts are used in the aqueous phase. This can be overcome by embedding $MoS_2$ within layered materials to achieve benefits to both materials, and improved performances can be realized [23,24]. $MoS_2$ is a photocatalyst for hydrogen evolution because it has a strong conductivity, wide basic surface region, sufficient band edge, solid lattice fit, and relatively high mobility. Layered $MoS_2$ is thought to be a good candidate for pairing with $g-C_3N_4$ because of its p-type conductivity. $g-C_3N_4$ is also an important photocatalytic substance for fuel generation from sunlight via water splitting and environmental remediation via organic pollutant degradation. ZnO, and CdS, which are also effective photocatalysts, have been used to alter the $MoS_2$ catalyst to improve its photocatalytic efficiency. In recent studies, $MoS_2$ has been widely explored in fields other than photocatalytic degradation. One major application that was studied is towards $H_2$ production [25]. $MoS_2/Ni_3S_2$/reduced graphene oxide electrocatalyst was studied for alcohol fuel cells [26], and on bacterial wound treatment [27]. The bacterial capturing and deactivating using $MoS_2$-based nanocomposites have also been studied.

Access to clean freshwater is a necessity for all human beings and, to a large extent, determines the quality of our lives. Pollution of the water bodies, particularly with industrial wastewater containing a variety of organic and inorganic pollutants, is an environmental issue worldwide [28]. According to the 2017 United Nations World Water Development report, 80% of wastewater is dumped into the environment without proper treatment [29]. The availability of various pollutants in water has serious health consequences for human health as well [30]. Further, the exploding population has placed an additional pressure on our diminishing freshwater resources [31]. Rapid industrialization and increasing agricultural activities have led to the generation of a large amount of wastewater [32]. Pollution of water also has adverse effects on the aquatic ecosystem. There is an urgent need to address this sorry state of affairs. Remediating wastewater through economic and eco-friendly protocols needs to be devised. One such technology for wastewater remediation is photocatalysis, which makes use of freely accessible sunlight to fuel the breakdown of organic contaminants present in wastewater [33]. Photocatalysis has risen in popularity in recent years because of its huge potential for resolving the growing energy crisis and environmental pollution. At moderate temperatures, solvothermal techniques can produce acceptable yields with varied controllability in the size of nanomaterials, which are successfully used in a variety of semiconductor applications [34].

This article briefly discusses the structure, geometry, methods of preparation, characterization, and application of $MoS_2$-based nanomaterials. The research achievements made in improving the degradation performance of $MoS_2$-based photocatalysts have been explored. The schematic illustration of the $MoS_2$-based catalysts' preparation, morphology, and applications are given in Figure 1.

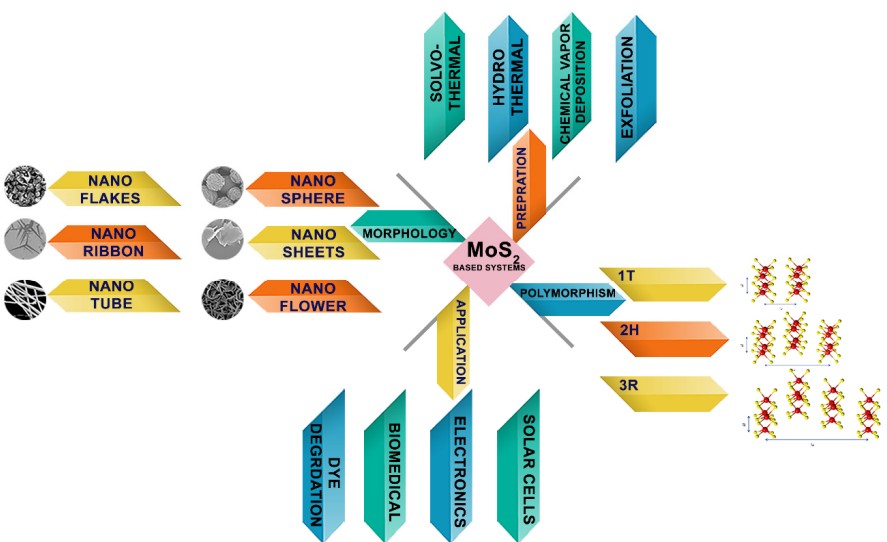

**Figure 1.** Representation of the preparation, morphology, and applications of MoS$_2$-based catalysts.

## 2. Structure and Geometries of MoS$_2$

MoS$_2$ is a black-coloured substance, insoluble in water, and like graphene, has a layered structure. It is mostly found in nature in the form of molybdenite MoS$_2$ has excellent thermal and chemical stability, which is a feature of layered transition metal compounds in general. As a result, it is employed in nanochemistry, catalysis, electrode materials, pharmaceuticals, nanomedical transportation, etc. MoS$_2$ is a typical n-type semiconductor with a layered structure that is quite similar to graphene. The band gap of MoS$_2$ grows as the number of the atomic layers decreases, resulting in desirable photoelectric characteristics. MoS$_2$ is a stable 2D transition metal dichalcogenide, which means it is a semiconductor of the MX$_2$ type, where M stands for transition metals and X is for chalcogen [29]. A single sheet is similar to a "sandwich" structure, where the Mo atom is sandwiched between two S atoms [35,36]. MoS$_2$ layers, separated by 0.65 nm, are held together by van der Waals forces, and the layers are bound together by strong covalent bonds [37]. Figure 2a,b show the crystal structure of MoS$_2$ [16,38,39].

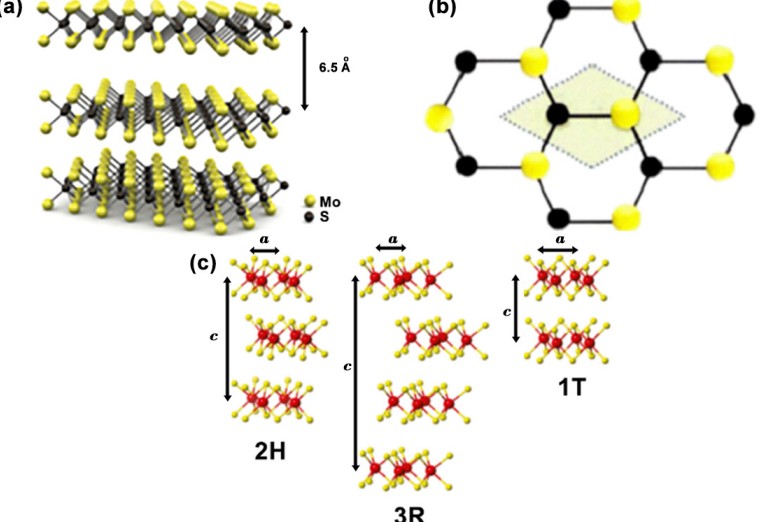

**Figure 2.** (**a**) Lateral view [16], (**b**) vertical view [38] of MoS$_2$ crystal structure, and (**c**) Crystal structure of MoS$_2$, 2H (left), 3R (centre), and 1T (right), respectively. Atom colour: Red-Mo, yellow-S [39]. Reprinted with permission from ref. [16,38,39]. Copyright 2011-Nature research, Copyright 2021-Elsevier, and Copyright 2016-Cell press.

MoS$_2$ is polytypic [4], with three distinct configurations: 1T [40], 2H [41], and 3R [42], all of which belong to symmetry point groups D$_6$h, D$_{6d}$, and C$_{3V}$, respectively [13]. The polymorphic structures are given in Figure 2c. The first digit in these polytypes denotes the number of layers in the make-up, while the alphabet denotes the crystallographic configuration. In these polytypes, the letters T, H, and R stand for trigonal, hexagonal, and rhombohedral arrangements. Various allotropes and morphologies, such as 3D (flowers, snowflakes, and dandelion patterns), 2D (nanosheets, nanostrips, and nanoribbons structure), 1D (nanowires and nanorods arrangement), and 0D (nanowires and nanorods) can be visualized, designed, and achieved by suitably altering the synthesis process. MoS$_2$ is a naturally occurring 2H molecule with 3% 3R that is thermodynamically stable.

MoS$_2$ monolayers show less activity in catalysis due to the limited charge mobility and a lack of metal edge sites. Increasing the number of exposed edge sites and using monolayered MoS$_2$ sheets having greater electrical conductivity have both improved the catalytic performances of MoS$_2$. However, the bulk of the metal on the basal plane of the monolayers still shows a lower catalytic activity [43]. From the studies conducted, it is evident that while the basal planes of 2H-MoS$_2$ are catalytically inert, the sulphur edge sites, and vacancy defects are the active sites for dye degradation. The photocatalytic dye degradation shows an enhancement when the size of the 2H-MoS$_2$ particles has been decreased to produce a higher density of edge sites. Another method for improving the catalytic activity of MoS$_2$ is to alter the 2H crystal phase to produce a polymorph with improved electrical conductivity, which will facilitate the transport of electrons to the active sites [44,45]. As stated below, MoS$_2$ comes in a variety of crystalline forms. The images given below are from the Materials projects database [46].

**Hexagonal:** The hexagonal structure of the MoS$_2$ is given in Figure 3. The structure of *P6$_3$/mmc* space group is given the Figure 3a. It shows a band gap of 1.465 eV and a density of 4.05 g/cm$^3$. Lattice parameters of the structure are a = b = 3.190 Å, and c = 14.879 Å. α = β = 90° γ = 120° and volume is 131.151 Å$^3$. The structure of *P6m2* is given in Figure 3b. It has a band gap of 1.661 eV and a density of 2.80 g/cm$^3$. Lattice parameters of the structure are a = b = 3.190 Å and c = 17.440 Å. α = β = 90° γ = 120° and volume is 153.721 Å$^3$ [46].

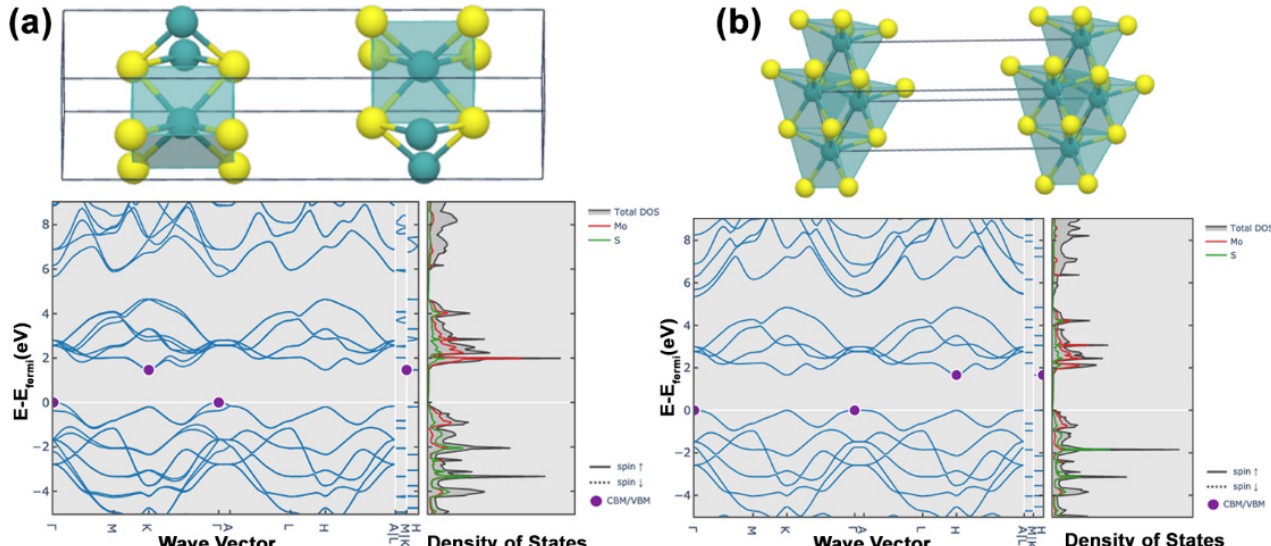

**Figure 3.** The geometrical representation of Hexagonal MoS$_2$ with their band structure and density of states of space group (**a**) *P6$_3$/mmc,* (**b**) *P6m2.* Reprinted with Creative Commons Attribution 4.0 License (CC BY 4.0) [46]. This work is licensed under copyleft, a CC BY 4.0 open source license.

**Trigonal:** The trigonal structure of the MoS$_2$ is given in Figure 4. The structure of the *R3m* space group is given in Figure 4a. It shows a band gap of 1.578 eV and a density of 4.24 g/cm$^3$. Lattice parameters of the structure are a = b = c = 7.341 Å. α = β = γ = 25.103°, and volume is 62.649 Å$^3$. The structure of MoS$_2$ with the *P3m1* space group is given in Figure 4b. It shows a band gap of 1.554 eV and a density of 2.42 g/cm$^3$. Lattice parameters of the structure are a = b = 3.190 c = 24.879 Å. A = β = 90° γ = 120°. Volume is 131.151 Å$^3$. The structure of the *P6m2* space group is given in Figure 4c, having a band gap of 1.509 eV and a density of 2.8 g/cm$^3$. Lattice parameters of the structure are a = b = 3.190 c = 32.319 Å. α = β = 90° γ = 120° and volume is 284.872 Å$^3$.

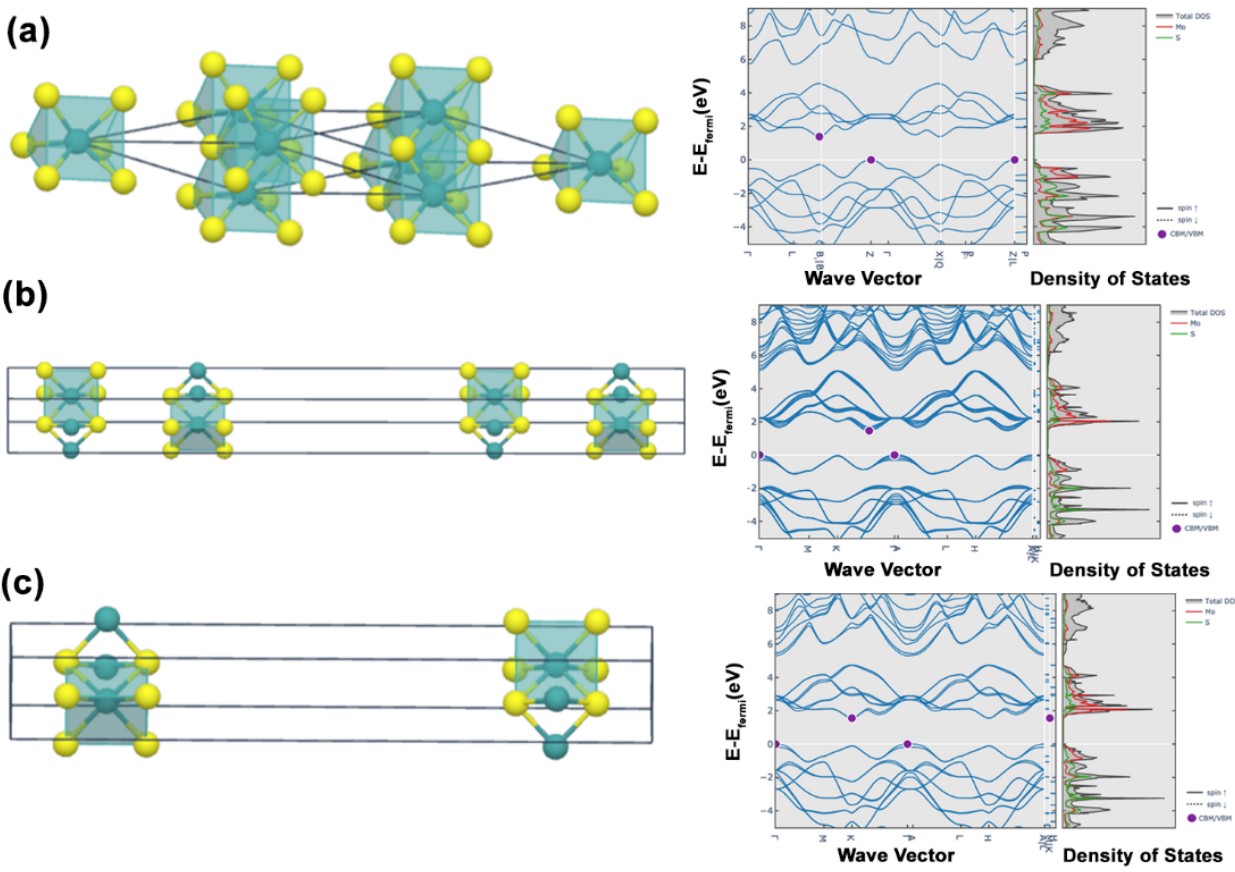

**Figure 4.** The geometrical representation of trigonal MoS$_2$ with their band structure and density of states of space group (**a**) *R3m,* (**b**) *P3m1,* (**c**) *P6m2.* Reprinted with Creative Commons Attribution 4.0 License (CC BY 4.0) [46]. This work is licensed under a copyleft CC BY 4.0 open source license.

**Other geometries:** The orthorhombic structure of the MoS$_2$ is given in Figure 5a. It has a band gap of 1.578 eV and a density of 2.67 g/cm$^3$. The structure belongs to the *Pmmn* space group. Lattice parameters of the structure are a = 3.163 b = 13.021 c = 43.774 Å. α = β = γ = 90°, volume = 1790.583 Å$^3$. Figure 5b represents the cubic geometry of MoS$_2$. This structure has a band gap of 1.562 eV and a density of 4.81 g/cm$^3$. The structure belongs to the *F43m* space group. Lattice parameters of the structure are a = b = c = 6.778 Å. α = β = γ = 60°, and volume = 221.129 Å$^3$. Figure 5c represents the tetragonal structure of MoS$_2$, and it has a band gap of 1.785 eV and a density of 3.27 g/cm$^3$. The structure belongs to the *I42d* space group. Lattice parameters of the above structure are a = b = c = 6.505 Å. α = β = 128.852°, γ = 75.248°, and volume = 162.514Å$^3$ [46].

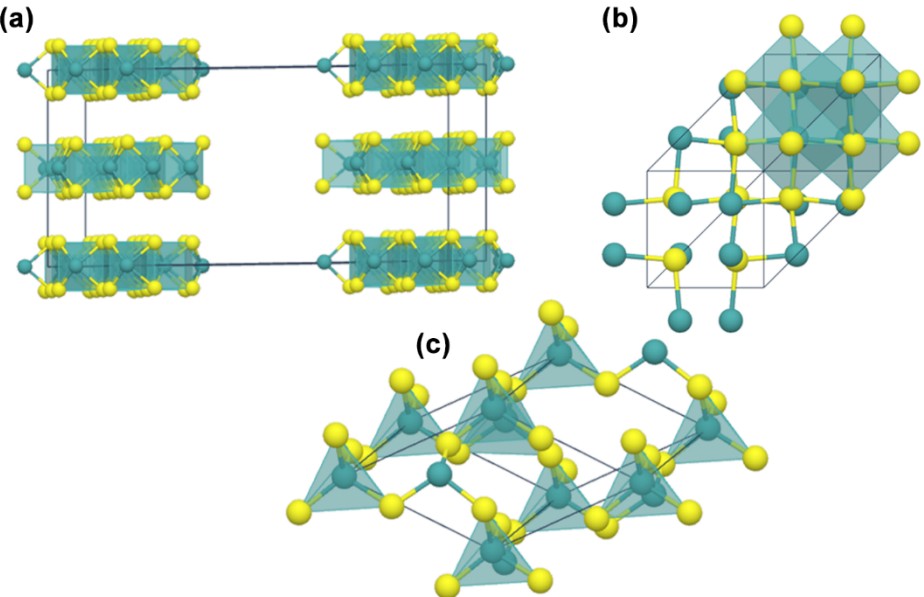

**Figure 5.** The geometrical representation of (**a**) Orthorhombic, (**b**) Cubic, (**c**) Tetragonal $MoS_2$. Reprinted with Creative Commons Attribution 4.0 License (CC BY 4.0) [46]. This work is licensed under a copyleft CC BY 4.0 opensource license.

## 3. Different Methods of Preparation of $MoS_2$

For the preparation of $MoS_2$, a variety of physical and chemical approaches have been used. The generally used physical methods are (a) mechanical exfoliation, (b) sputtering, (c) epitaxy, (d) plasma, etc. The lattice structure of the material will not change or be destroyed in this method, provided the precursor materials are pure [38]. However, in recent studies, physical methods are not commonly used since they are not cost effective. The chemical methods include (a) chemical vapor deposition method, (b) hydrothermal method, and (c) solvothermal method. The most common methods of preparation are listed in Figure 6.

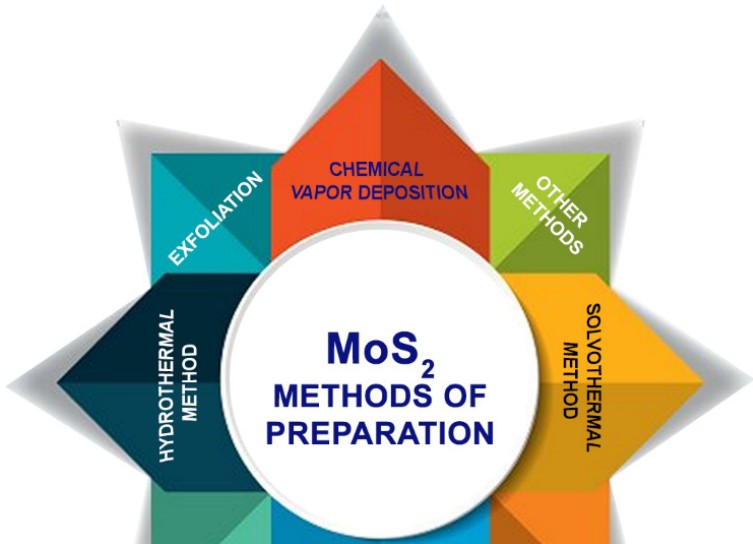

**Figure 6.** Different methods of preparation of $MoS_2$-based systems.

### 3.1. Physical Methods

**Mechanical exfoliation:** Mechanical exfoliation is an approach used in the preparation of 2D materials, similar to the method used in monolayer graphene. Yu et al., developed a simple, environmentally friendly, and scalable exfoliation approach for water-dispersible

MoS$_2$ [47]. In the pre-treatment of MoS$_2$ powder, no intercalation agents have been used. Then, dispersion took place in ultrasound-assisted water—a typical exfoliation process for nanosheets—in combination with viscoelastic stinging for the preparation of MoS$_2$. Funke et al. used an micromechanical exfoliation method where they did not find any evident grid defects in elliptic-polarization spectra imaging [48]. This method is useful to manage the materials' surface and mechanical exfoliation. One drawback here is that the thickness and transverse dimensions are difficult to regulate and, therefore, the procedure is unsatisfactory and unsuitable for commercial manufacturing [49].

### 3.2. Chemical Methods

***The chemical vapor deposition (CVD) method:*** This method can be used to prepare MoS$_2$ and transition metal dichalcogenides of high quality on a large scale, mainly to make MoS$_2$ films [50,51]. It is a bottom-up approach that preserves the defects, crystallinity, and morphology of the material [52]. CVD is a large-scale chemical reaction that involves vapors reacting with the substrate to produce thin films. Direct evaporation is used in the CVD process, commonly known as the vapour solid development technique. It creates a high-quality monolayer with fewer tiny flakes on the substrate. Precursors are charged and sublimated into a gaseous state which helps to undergo chemical reactions at high temperatures. Strong sediments grow on the matrix surface during condensation; hence it is commonly used to prepare monolayer nanocomposite systems and multilayer transition metal MoS$_2$ with good control on the number of layers.

MoO$_3$ powder is used as a molybdenum source and S powder as a sulfur source in the preparation of large-area monolayer MoS$_2$ films. Monolayer films grow consistently with a uniform nature, but defects created during the growth process cause a reduction in the material consistency [53]. In the preparation, improvements were made by Zhang et al., who produced triangular monolayers of MoS$_2$ with grain sizes up to 150 μm [54]. Alharbi et al. used a chemical vapor deposition process to create a large-area MoS$_2$ film with the highest island density in the field [55]. Balendhran et al. developed a two-step bulk fabrication technique using an MoO$_3$ precursor [56]. In a quartz channel, MoO$_3$ is evaporated on every substrate, followed by sulfidation. A schematic representation of synthesizing MoS$_2$ using the chemical vapour method and CVD growth of the MoS$_2$ when Mo foil is used as a precursor is depicted in Figure 7a,b, respectively [57,58].

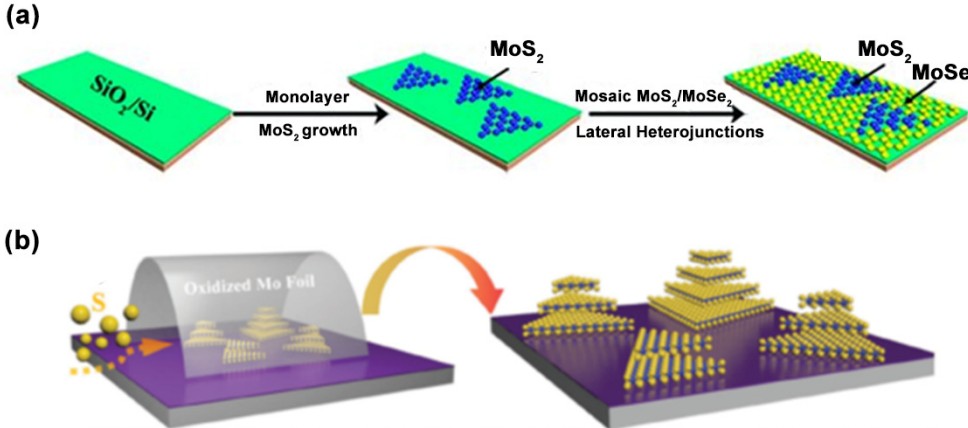

**Figure 7.** (**a**) Representation diagram of the synthesis of MoS$_2$ by chemical vapor deposition method [57] (**b**) Representation for the CVD growth of MoS$_2$ using arched oxidized Mo foil as the precursor [58]. Reprinted with permission from ref. [57,58]. Copyright 2018-MDPI, Copyright 2020-Elsevier.

**Hydrothermal method:** The hydrothermal method is a promising wet chemical method that is performed using a hydrothermal bomb. The temperature used for this method is usually 180 °C [59]. Typically, the precursors used are sodium molybdate

($Na_2MoO_4 \cdot 2H_2O$) and ammonium molybdate tetrahydrate (($NH_4$)$_6Mo_7O_{24} \cdot 4H_2O$) for molybdenum and thiourea ($CS(NH_2)_2$) for sulfur. Nanosheets [60], nanospheres [61], nanoflowers [62], and nanotubes [63] can be prepared using hydrothermal methods. It is the most frequently used method for synthesis of flower-like $MoS_2$ [64]. Figure 8 represents the preparation steps of $MoS_2$ nanosheets. The molybdenum source was $Na_2MoO_4 \cdot 2H_2O$, and the sulfur source was thiourea in the synthesis proposed by Zou et al. [65]. The $MoS_2$-RGO-3 was generated using a hydrothermal process and had a structure that resembles a cabbage, with distinct lattice fringes and a diameter range of 500 nm$^{-1}$. 3D $MoS_2$ ultrastructure was prepared by Anwer et al. [66], who used a controllable hydrothermal approach to achieve their results. These micro-sized marigold flower-like patterns were made up of 2D ultrathin $MoS_2$ nanosheets that were randomly spaced but closely connected. Controlling the input concentrations of thiourea and the precursor of $MoS_2$ resulted in the $MoS_2$-microflower composition. Tong et al. [67] have used a previously published hydrothermal method for $MoS_2$ preparation. The flower-like $MoS_2$ had a hierarchical structure with a diameter of 500 nm and was subjected to an 18 h hydrothermal reaction at 220 °C. The $MoS_2$ nanoflowers made by the hydrothermal process were composed of uniform crystallinity and size with minimum agglomeration. Zhou et al. [68] have replaced the sulfur source with thioacetamide in their studies. Xia et al., in their recent study, synthesized the CdS@$MoS_2$ nanocomposite using the hydrothermal method, and evaluated the photocatalytic degradation of Rhodamine 6G (Rh 6G), and found that adsorption in the nanocomposite increases when the $MoS_2$ amount is increased, showing a great efficacy in processing Rh 6G [69]. Jian et al. used a two-step hydrothermal method to prepare the nanocomposite $MoS_2$/BiOBr, which is very efficient in Rhodamine B (RhB) dye degradation [70].

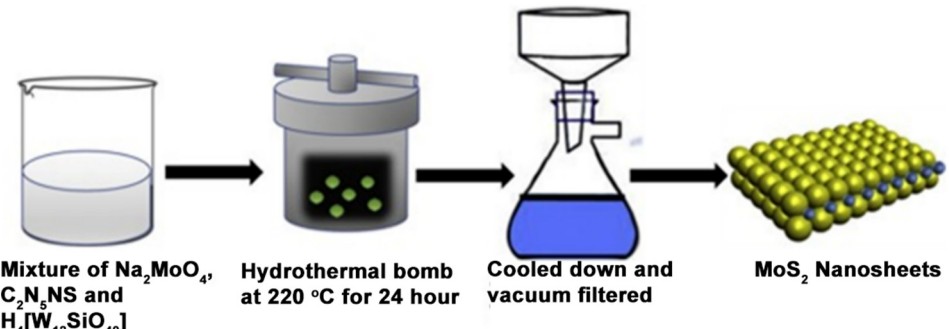

**Figure 8.** Schematic representation of synthesis of $MoS_2$ using hydrothermal method [60]. Reprinted with permission from ref. [60]. Copyright 2018-Elsevier.

Additionally, the regulating parameters under hydrothermal parameters are: (a) the percentage of components, (b) the reaction temperature, (c) the solution, (d) the pH value, and (e) the solution concentration. These variables might be effectively controlled to monitor the chemical reaction and the shape of the substance prepared. $MoS_2$ shows enhanced photothermal properties when synthesized using other processes such as hydrothermal process, a photo-deposition technique, and an in situ solid-state chemical reduction approach. Moreover, at low temperatures, the Ag/$MoS_2$/$TiO_2$-x composite had outstanding degradation properties. Its hydrogen production rate was considerably higher than that of pristine $TiO_2$ [71].

**Solvothermal method:** The solvothermal and hydrothermal methods of syntheses are similar techniques [72]. An organic solvent is used in the latter technique instead of water. Thus, where hydrothermal method cannot be applied for compounds that are very easily hydrolyzed, the solvothermal method can be used. In the solvothermal process, non-aqueous solvents are utilized as pressure carriers, intermediates, and mineralizers. Simultaneously, several non-aqueous solvents with a variety of characteristics can be utilized, resulting in nanomaterials with a diverse set of properties. Solvothermal processes

can be used to make core-shell composites. Bai and collaborators generated a $Co_9S_8@MoS_2$ core-shell heterojunction via a solvothermal method, which enhanced the catalytically active sites [73]. Zhang et al. used a basic solvent thermal approach to make $MoS_2$/carbon nanotubes with a core-shell structure having improved optical properties [74]. To build $Fe_3O_4@SiO_2@MoS_2/g\text{-}C_3N_4$ (FSMG), Lu et al. employed a unique and efficient method where the $Fe_3O_4$ spheres formed the inner core of the $SiO_2$ shell—placed using the sol-gel process—and were made using the solvothermal method. FSMG demonstrated good degradation behaviour against RhB when exposed to visible light. Importantly, by adjusting the proportion of water in the solvent, the size of the gel can be easily managed. Gao et al. have reported a solvothermal preparation by mixing $MoS_2$ from ammonium tetrathiomolybdate with dimethylformamide (DMF) and hydrazine [75]. The mixture was then autoclaved for 10 h with 200 °C. The preparation of $MoS_2$ aggregate is given in Figure 9.

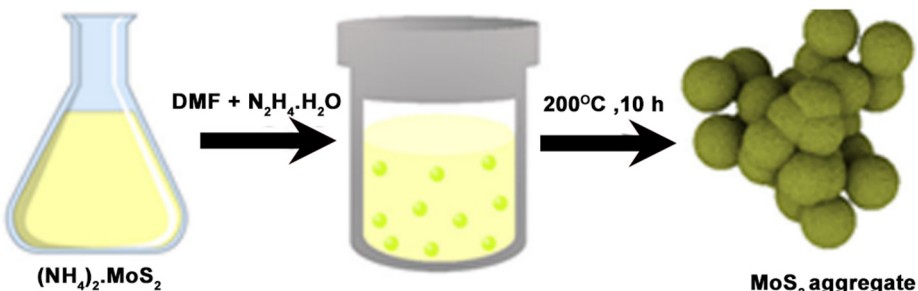

**Figure 9.** Schematic diagram of solvothermal synthesis of $MoS_2$ nanosheet aggregates [75]. Reprinted with permission from ref. [75]. Copyright 2018-Nature research.

**Other methods:** The methods discussed below are some of the most novel ones, namely, (a) thermal-based sulfurization process, (b) ultrasonic-assisted homogenous magnetic stirring method, (c) ultrasound-assisted cracking system, (d) thermal evaporation technique-based systems, (e) in situ growth method, (f) hydrolysis method, and (g) dual-template approach. Hollow structures may be developed using the dual-template approach, where the method increases the active sites and characteristics of produced $MoS_2$ nanosheets [76]. Thermal sulfurization has been used to create homogeneous $MoS_2$ films [77]. This approach does not require any pre-processing, unlike chemical vapour deposition (CVD). Acid treatment and ultrasonic irradiation are required before the ultrasound-assisted cracking procedure in order to obtain ideal materials [78]. Hydrolysis is a process for preparing nanoparticles in which precursor molecules are appropriately hydrolyzed in an aqueous solution environment under specific conditions. Zhu et al. reported the synthesis of $MoS_2$ nanosheets of uniform thickness using a quick and fast $LiBH_4$ hydrolysis reaction [79].

## 4. Characterization of $MoS_2$-Based Materials

### 4.1. Morphological Properties

The shape of $MoS_2$-based photocatalysts affects their efficiency. Depending on the technique of preparation, $MoS_2$ can have a variety of morphologies. Jaleel et al. created $MoS_2$ with a nanoflower shape utilizing a hydrothermal procedure using ammonium molybdate as the Mo source and thiourea as the S source [59]. This nanocomposite was used for degrading the malachite green dye with great efficiency. The SEM image of the nanoflower $MoS_2$ is given in Figure 10a [59]. Sun et al. used the CVD process to create nanosheets of $MoS_2$ and the hydrothermal intercalation method to construct nanospheres and used them for the adsorption studies. Figure 10b represents the nanosphere structure of $MoS_2$ [80]. Visic et al. and co-workers prepared $MoS_2$ coaxial nanotubes by the sulfurization transition of M6S2I8 nanowires in an argon environment with a gas flow of $H_2/H_2S$ mixture. They proposed this $MoS_2$ as an analogous material to graphene. Figure 10c represents the nano-wired $MoS_2$ [81]. Mahyavanshi et al. used a chemical vapour deposition technique to

illustrate the directional growth of $MoS_2$ monolayer ribbons (Figure 10e) in a sulfur-rich environment. An array of molybdenum and sulphur edge terminations with 60° and 120° angle formations are present in this nanocomposite, as evidenced from the SEM and TEM images. There may be new opportunities for electrical and electrochemical applications due to the directed development of $MoS_2$ ribbons with specified edge structures under specific CVD conditions [82]. The $MoS_2$ nanoflakes were synthesized by Wu et al., (Figure 10f) via effective grinding-assisted sonication exfoliation for photoluminescence studies of $MoS_2$ [83]. SEM images of nanoflower, nanosheet, nanosphere, nanotube, nanoribbon, and nanoflake morphologies are depicted in Figure 10a–f, respectively [83]. The HRTEM images in Figure 11a,b shows a close interface between the three components, which could assist with electron transfer to the composite surface [84,85].

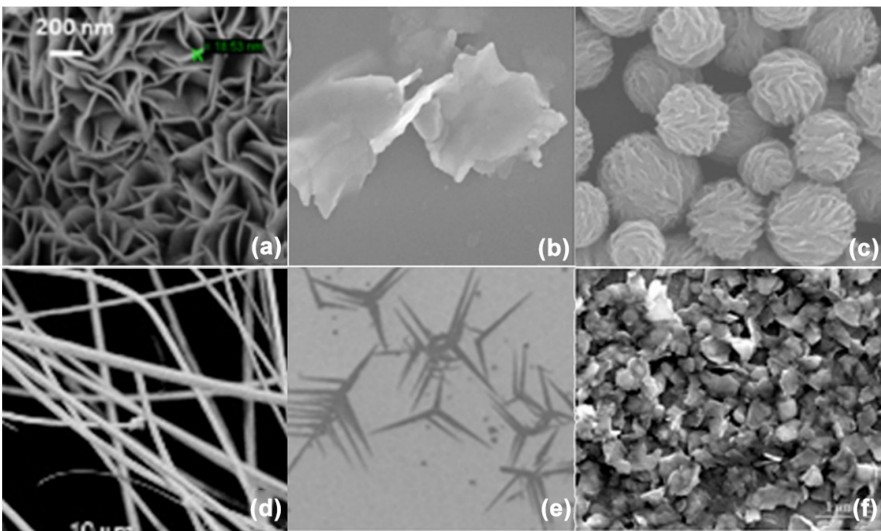

**Figure 10.** (**a**) Nanoflower [59], (**b**) Nanosheets [80], (**c**) Nanosphere [80], (**d**) Nanotube [81], (**e**) Nanoribbon [82], (**f**) Nanoflakes [83]. Reprinted with permission from ref. [59,80–83]. Copyright 2021-Springer, Copyright 2015-OSA publication, Copyright 2011-Springer, Copyright 2017-Elsevier, Copyright 2014-Hindawi publication.

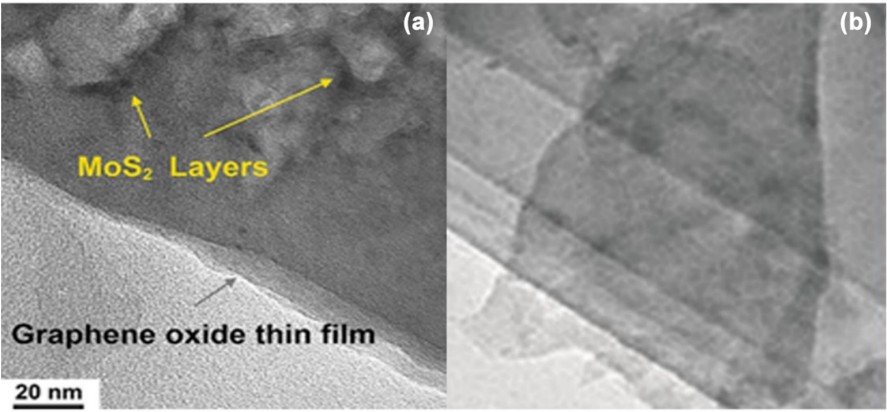

**Figure 11.** (**a**,**b**) TEM images of $MoS_2$-based system. Reprinted with permission from ref. [84,85] 2015-Nature research, Copyright 2013-Springer.

### 4.2. Structural Properties

The various characterization techniques used to determine the structure of $MoS_2$ are discussed here [59,81,86–88].

X-ray diffraction (XRD) has a significant role in identifying the crystalline nature of the material in nanocomposite studies. $MoS_2$ shows different peaks in the XRD as shown in

Figure 12a. The major plane obtained is (002) at 14.38° [89]. This corresponds to about eight layers of MoS$_2$. Besides this major peak, MoS$_2$ also displays peaks that correspond to (100), (101) at 32°, (103) at 40°, (105) at 49°, (110) at 58.5°, (008), and (200) planes [81]. There is no peak corresponding to (001) plane as there is no MoO$_3$ [90]. Infrared spectral analysis (FTIR) is the other major characterization tool used to identify the functional groups present in the nanomaterial. As shown in Figure 12b, the FTIR plot of MoS$_2$ shows various peaks. The peak at 3182 cm$^{-1}$ is due to the O−H group, while 639 cm$^{-1}$, 893.39 cm$^{-1}$, 1402.99 cm$^{-1}$, and 1622.8 cm$^{-1}$, are the broad absorption bands attributed to MoS$_2$. And the band at 483.23 cm$^{-1}$ and 931.39 cm$^{-1}$ are due to S−S bond [88].

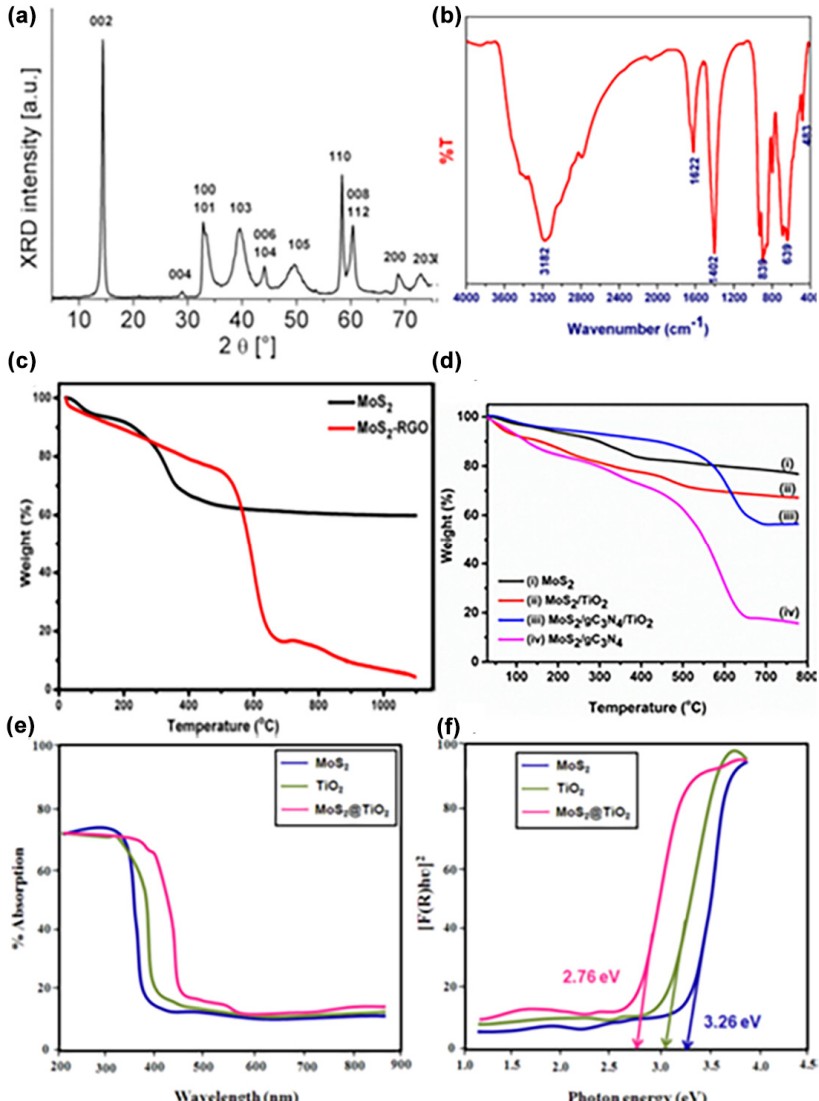

**Figure 12.** (**a**) XRD plot of MoS$_2$, (**b**) FTIR plot of MoS$_2$, (**c,d**) TGA curves of MoS$_2$, (**e**) UV-DRS graph of MoS$_2$, (**f**) Tauc plot of MoS$_2$. Reprinted with permission from ref. [59,81,86–88]. Copyright 2011-Springer, Copyright 2019-Springer. Copyright 2020-Springer, Copyright 2017-American Chemical Society. Copyright 2020-Springer.

Thermogravimetric analysis (TGA) on the MoS$_2$ shows the thermal stability of the material. The TGA plot of MoS$_2$ is given in Figure 12. In Figure 12c,d, we can see a similar pattern of thermal decomposition of the MoS$_2$. Bahuguna et al. [87] synthesized MoS$_2$ using a hydrothermal process and then carried out TGA analysis. MoS$_2$ shows an initial loss of mass at about 100 °C, which is caused by the evaporation of adsorbed water molecules, as seen in Figure 12c. Further, MoS$_2$ loses mass at around 300 °C, due to the decomposition of

sulfur from the compound, and the $MoS_2$ oxidizes to $MoO_3$, after which it remains stable up to 1000 °C. Figure 12d(i) shows the TGA analysis of the $MoS_2$ and its modified system [59].

### 4.3. Optical Properties

The optical properties are usually characterized by two different characterizing tools (a) photoluminescence (PL) and (b) UV-diffuse reflectance spectroscopy (UV-DRS) study. From the UV-DRS analysis, and Tauc plot, the band gap is calculated using Kubelka–Munk function, which is given in Figure 12e,f. The PL spectrum has an important role in proposing a mechanism in organic dye degradation using nanocomposites. The band gap of $MoS_2$ using Tauc plot has been calculated for $MoS_2$ as 3.26 eV [86]. Zhang et al. prepared a ternary system of $MoS_2/g$-$C_3N_4/TiO_2$ and calculated the band gap as 2.76 eV. Chakrabarty et al. determined the band gap of RGO-$MoS_2$ supported $NiCo_2O_4$ as 2.36 eV [91]. Nayak et al. prepared $MoS_2/NiFe$ layered double hydroxide (LDH), and using Tauc calculation, found 1.86 eV as the band gap [92]. In the modified $MoS_2$ system, the UV-DRS of the composites displayed a red shift towards the visible light spectrum [93].

## 5. $MoS_2$-Based System for Degradation of Organic Dyes

Because of its strong optical absorption, $MoS_2$ has a wide range of applications in photochemistry, photocatalysis, and photoelectronic research. In electronics, the $MoS_2$-based materials are used as photodetectors [94], field-effect transistors [95], solar cells [96], chemical sensors [97], and so on. $MoS_2$ is also used in the transportation of drugs in cancer chemotherapy. Major applications of $MoS_2$ and $MoS_2$-based systems in various photocatalytic degradation reactions are discussed here.

### 5.1. Binary Systems of $MoS_2$

Graphitic carbon nitride (g-$C_3N_4$) has been thoroughly explored for its small band gap (2.7 eV), nonmetal characteristics, nontoxicity, ease of availability, and excellent thermal and chemical stability since its debut in 2009 for photocatalytic hydrogen evolution. Many studies have been done by fabricating $MoS_2$ on g-$C_3N_4$. Zhang et al. [98] prepared a nanocomposite by fabricating $MoS_2$ and g-$C_3N_4$. Here, $MoS_2$ was prepared by the hydrothermal method. The photocatalytic efficiency of the samples was assessed by measuring their ability to degrade the organic RhB and methyl orange (MO) dyes in aqueous solutions when exposed to visible light. The prepared catalysts showed a satisfactory amount of degradation of these dyes. Benavente et al. [99] synthesized a catalyst $ZnO/MoS_2$ by a sol-gel method and investigated the degradation of 10 ppm of Methylene blue (MB) dye using visible light for 300 min, resulting in a 75% degradation.

Wang and their colleagues have prepared a binary system of $MoS_2$ with $TiO_2$. The $MoS_2$ is synthesized using a simple hydrothermal method, and their ability to degrade RhB, MB, and MO dyes for 60 min showed activities of 99.6, 96.4, and 87.4%, respectively [100]. Selvaraj et al. [101] have prepared a type-II $MoS_2/ZnO$ composite via the hydrothermal method. It degraded the MB dye using UV light and demonstrated 99% degradation activity. In a recent study by Bargozideh et al. [102], $MoS_2$ nanoflowers were made using a hydrothermal process and fabricated with $BiFeO_3$, which showed degradation of 89% on RhB dye.

Ding and co-workers have modified $MoS_2$ with graphene oxide (GO), and studied for the degradation of MB dye [103]. A total of 10 mg of the nanocomposite was used to study the degradation of MB dye in the presence of visible light. The composite showed a degradation of 99%, which is an enhancement of 20% activity over the pristine $MoS_2$. In this study, they have synthesized $MoS_2$ using solvothermal method using n-butyl lithium as solvent. The preparation of GO was carried out by the sol-gel method. Tang et al. [104] in a recent study, prepared modified $MoS_2$ using $SrZrO_3$ via the hydrothermal method and studied the photodegradation of the MB dye using visible light irradiation. The degradation efficacy was found to be 99.7% for 15 mg nanocomposite in 25 ppm of MB dye. $MoS_2/ZnS$ synthesized via hydrothermal method have been used for the degradation of MB dye. In total, 10 ppm dye solution along with 50 mg of nanocomposite gave

99.89% degradation activity [105]. In a recent study, Jian et al. constructed a BiOBr loaded $MoS_2$ and studied the photocatalytic activity on the RhB dye [70]. Using 20 mg of nanocomposite in 10 ppm solution of RhB dye in Xenon arc light achieved 96% efficiency. Liu et al., synthesized a flower-like $CeO_2/MoS_2$ composite and studied its catalytic activity under visible light for the degradation of MO dye. The nanocomposite showed a 96.1% efficiency in 90 min [106]. Sharma et al., used a biosynthetic approach to construct a 3D/2D $CeO_2/MoS_2$ nanocomposite [107]. Here, 20 mg of the composite decomposed 20ppm of Methyl violet (MV) dye solution under a visible light source (300 W). The efficiency was found to be 96.25% in 90 min. Wang et al. [108], synthesied a $CeO_2/MoS_2$ 2D nanostructure via hydrothermal method for the reduction study of aqueous Cr(VI). Then, 6 mg of catalyst and 5ppm of Cr(VI) resulted in a 99% removal of Cr(VI) in 120 min.

From all these studies, it can be seen that the fabrication of effective photocatalysts on $MoS_2$ reduces the band gap of the final composite prepared. All of the photocatalysts discussed below are semiconductor photocatalysts, where the narrowing of the band gap considerably boosts catalytic activity.

### 5.2. Ternary Systems of $MoS_2$

Different oxides have been doped with $MoS_2$ to increase their degradation efficacy in photocatalysis. g-$C_3N_4$ is a metal-free semiconductor having a layered structure with great electron-proton transferability and has found applications as a photocatalyst [109]. Incorporating $TiO_2$ into the binary system above to obtain $MoS_2$/g-$C_3N_4$/$TiO_2$ ternary system has led to a significant increase in the action of $MoS_2$ and g-$C_3N_4$ photocatalysts.

Jo et al. synthesized a ternary $MoS_2$/g-$C_3N_4$/$TiO_2$ system in 2015 [110]. The composite was prepared over multiple steps. Initially, $MoS_2$ was prepared using hydrothermal method and then exfoliated with g-$C_3N_4$. $TiO_2$ was added by the impregnation method to obtain $MoS_2$/g-$C_3N_4$/$TiO_2$. In the photocatalytic experiment, to 10 ppm MB, 30 mg of prepared catalyst was added, and the reaction observed for 60 min under visible light irradiation. The system g-$C_3N_4$ (10%)/$TiO_2$/$MoS_2$ (0.2%) showed an activity of 99.5%. Atrazine herbicide degradation has also been studied under the same conditions for 300 min to obtain 86% degradation [110]. In 2016, Zhang et al. [111] fabricated $TiO_2$/g-$C_3N_4$/$MoS_2$ composite from g-$C_3N_4$ and exfoliated it with hydrothermally synthesized $MoS_2$ to obtain $TiO_2$/g-$C_3N_4$/$MoS_2$ using solvothermal method and used for MO degradation (20 ppm of MO solution). The degradation of pure $TiO_2$, pure g-$C_3N_4$, g-$C_3N_4$/$MoS_2$, and $TiO_2$/g-$C_3N_4$ was 28.9, 22.4, 26.7, and 90.1%, respectively (100 mg catalyst). In comparison, all $TiO_2$/g-$C_3N_4$/$MoS_2$ composites had better photocatalytic activity than pure $TiO_2$, g-$C_3N_4$, and $TiO_2$/g-$C_3N_4$ composites, demonstrating that $TiO_2$ and g-$C_3N_4$/$MoS_2$ hybrid had a synergistic impact. The Gaussian09 programme was used to perform DFT investigations on the as-prepared composite photocatalyst for both geometry optimization and frequency estimation [111]. Figure 13 shows the optimized $TiO_2$/g-$C_3N_4$/$MoS_2$ composite structure utilizing the LANL2DZ basis package at the density functional B3LYP level. Following the calculations of the $TiO_2$, $MoS_2$, g-$C_3N_4$, and $TiO_2$/$MoS_2$ moieties, the structures of potential $TiO_2$/g-$C_3N_4$/$MoS_2$ combinations were computed using the B3LYP density functional theory with the LANL2D basis set. The B3LYP-D3 method was used to optimize the adsorption energy of a $TiO_2$/g-$C_3N_4$/$MoS_2$ composite.

In 2020, Mahalakshmi et al. reported the g-$C_3N_4$/$MoS_2$/$TiO_2$ system for photodegradation [112]. The g-$C_3N_4$, synthesized from melamine was added to titanium isopropoxide (TTIP) with rapid stirring. Then the hydrothermally prepared $MoS_2$ and the above-prepared solutions were added, and hydrothermal synthesis was carried out to obtain a $C_3N_4$/$MoS_2$/$TiO_2$ system. They investigated the catalyst's activity at various weight percentages as well as the degradation of the MO dye. In 0.355 g of MO dye, 50 mg of catalyst was added. Under visible light irradiation, the reaction was carried out for 60 min. The catalyst showed an 88% degradation. The same hybrid catalyst has been used for the degradation of 4-nitrophenol (4-NP) to obtain a degradation efficiency of 74%. In a recent study, Jaleel et al. prepared the nanocomposite $MoS_2$/g-$C_3N_4$/$TiO_2$ using hydrothermal

and exfoliation methods [59]. Here, 50 mg of prepared nanocomposite was added to 50 ppm of malachite green (MG) dye to obtain 97% efficiency within 60 min. The Box–Behnken design of the response surface approach is also used to examine the optimal experimental conditions for dye degradation [59].

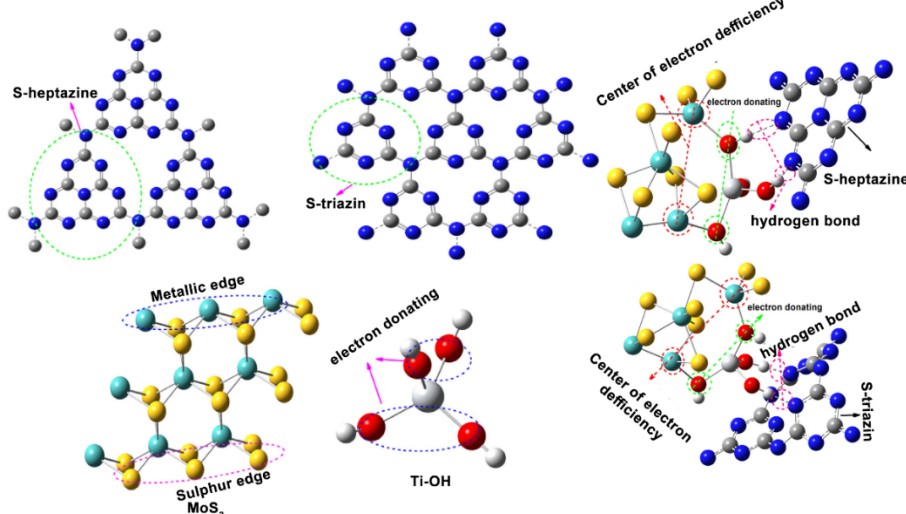

**Figure 13.** The optimized structure of $TiO_2/g$-$C_3N_4/MoS_2$ nanocomposite. (Blue-N, Red-O, White-Ti, Grey-C, Yellow-Mo, Cyan-S) Reprinted with permission from ref. [111]. Copyright 2018-Elsevier.

Modification of $MoS_2$ nanoparticles with the g-$C_3N_4$ and ZnO is also reported. Since the two photocatalysts are well-matched with overlapping band alignments, coupling ZnO with g-$C_3N_4$ could yield an excellent heterostructure where increased charge separation takes place [113]. Theoretically, electrons excited from the valence band (VB) to the conduction band (CB) of the g-$C_3N_4$ will then migrate to the CB of ZnO, such heterostructures have been shown to have enhanced photocatalytic activity. The $MoS_2/g$-$C_3N_4/ZnO$ nanocomposite is used to degrade the MB dye for 60 min, and the MB dye shows more than 90% degradation. Lu et al. [114] synthesized g-$C_3N_4$ and Ag on $MoS_2$ and investigated the degradation activity on the RhB dye for 90 min using a Xenon arc lamp as a light source. The amount of $MoS_2$ used in this analysis was 100 g, and the dye concentration was 20 ppm to yield a degradation of 95.8%. The $MoS_2$ has a flower-like morphology and was synthesized using a hydrothermal process.

Beyhaqi and colleagues prepared a nanocomposite by modifying the $MoS_2$ using g-$C_3N_4$ and $WO_3$. The photodegradation study was performed on three dyes RhB, MO, and MB using Xenon lamps as the light source. The degradation efficiency obtained was 99.9, 83.4, and 91.8%, respectively, for RhB, MO, and MB. The same nanocomposite could degrade ciprofloxacin drug completely within 2 h of irradiation of Xenon arc as light source [115]. Pant et al. [116] modified $MoS_2$ using CdS and $TiO_2$, and it was used to examine the breakdown of MB dye on carbon nanotubes. The dye degraded completely within 15 min of irradiation with visible light. Vignesh et al. modified $MoS_2$ using g-$C_3N_4$ and $Bi_2O_3$ and studied the reaction of MB degradation. When the visible light was irradiated for 90 min on the dye (20 ppm) using 50 mg catalyst, it gave a good degradation activity of 98.5% [117]. This nanocomposite was also used for the destruction of *E. coli* and *S. aureus* bacteria. Reusability plays an important role in catalysis. Most of the reports related to $MoS_2$-based systems have shown good reusability up to five cycles [59,116,118–120]. Talukar et al., synthesized a ternary hybrid nanoflower $CeO_2$-$ZrO_2@MoS_2$ to study the sonophotocatalytic degradation of naproxen (NPX) using a visible light source (250 W) LED lamp. In total, 50 ppm of NPX solution yielded an enhanced degradation of 96% in 40 min.

A summary of the $MoS_2$ nanocomposites and the degradation of various dyes has been given in Table 1 [121–141]. In all these studies, we see that the fabricating/doping agent

changes the degradational activity of the nanocomposite. The band gap shows a red shift in the modified $MoS_2$, which helps in the better utilization of visible light. The morphology and degradation activities have a direct relationship which in turn depends on the preparation technique. Many authors have done computational and statistical studies to understand the highest degradation efficacy and optimal conditions for the degradation of organic dyes.

**Table 1.** Comparison of the photocatalytic degradation of various dye using $MoS_2$-based catalysts.

| System | Dye | Reaction Conditions | | | | Degradation (%) | Ref. |
|---|---|---|---|---|---|---|---|
| | | Time (Min) | Light Source | Catalyst Amount (mg) | Conc of Dye (ppm) | | |
| $MoS_2/SnO_2$ | MB | 90 | Visible | 20 | | 95.0 | [30] |
| MoS2 nanosheets | MB | 120 | Fluorescent lamp | 10 | 100 | 49.3 | [121] |
| $CeO_2$-$ZrO_2$@$MoS_2$ | NPX | 40 | Visible light | | 50 | 96.0 | [142] |
| $CeO_2/MoS_2$ | MO | 90 | Visible light | 25 | 20 | 96.1 | [106] |
| 3D/2D $CeO_2/MoS_2$ | MV | 30 | Visible light | 20 | 20 | 96.25 | [107] |
| Cerium-doped $MoS_2$ | Cr(IV) | 30 | Visible light | 6 | 20 | 40 | [108] |
| $MoS_2/BiOBr$ | RhB | 30 | Xenon arc | 20 | 10 | 96.0 | [70] |
| $MoS_2$-x nanosheet arrays | RhB | 60 | Visible | | 5 | 97.2 | [122] |
| 10% g-$C_3N_4$/$TiO_2$/$MoS_2$(0.2) | MB | 60 | Visible | 30 | 10 | 99.5 | [110] |
| Layered $MoS_2$ | MB | 90 | Visible | 50 | 10 | 71.0 | [133] |
| $TiO_2/SnS_2/MoS_2$ | MB | 90 | Visible | | 5 | 81.8 | [135] |
| $TiO_2$/g-$C_3N_4$/$MoS_2$ | MO | 60 | Visible | 100 | 20 | 90.6 | [111] |
| g-$C_3N_4$/$MoS_2$/$TiO_2$ | MO | 60 | Visible | 50 | 0.03 | 88.0 | [112] |
| $MoS_2$/g-$C_3N_4$/$TiO_2$ | MG<br>RhB | 60 | Visible | 50 | 10 | 86.0<br>99.6 | [59] |
| $TiO_2/MoS_2$ | MB<br>MO | 60 | Visible | 50 | 10 | 96.4<br>87.5 | [100] |
| RGO-$MoS_2$ supported $NiCo_2O_4$ | RhB | 90 | Visible | 50 | 10 | 95.0 | [91] |
| $SrZrO_3$/Flower-like$MoS_2$ | RhB | 80 | solar | 15 | 25 | 99.7 | [104] |
| g-$C_3N_4$/Ag/$MoS_2$ | RhB | 90 | Xenon arc | 100 | 20 | 95.8 | [114] |
| $MoS_2$/NiFe LDH | RhB | 120 | Solar | 20 | 20 | 90.0 | [92] |
| $MoS_2$/COF | RhB | 30 | Solar | 10 | 20 | 98.0 | [136] |
| $MoS_2$–GO | MB | 60 | solar | 10 | 10 | 99.0 | [103] |
| $BiPO_4$, $MoS_2$ and graphene | RhB<br>RhB | 90 | Mercury | 100 | 5<br>50 | -<br>99.9 | [137] |
| g-$C_3N_4$/$WO_3$/$MoS_2$ | MO<br>MB | | Xenon arc | 100 | 20<br>20 | 83.4<br>91.8 | [115] |
| $MoS_2$/CdS/$TiO_2$ | MB | 15 | Visible | 25 | 10 | 100 | [116] |
| g-$C_3N_4$-based $MoS_2$ and $Bi_2O_3$ | MB | 90 | Visible | 50 | 20 | 98.5 | [117] |
| $MoS_2$/ZnS | MB | 32 | Visible | 50 | 10 | 99.9 | [105] |
| $MoS_2$/$CdIn_2S_4$ | RhB | 30 | Visible | | 10 | - | [138] |
| $CoFe_2O_4$/$MoS_2$ | CRMBMO | 60 | Visible | 30 | 20 | 94.9<br>67.8 | [139] |
| ZnO/$MoS_2$ | MB | 300 | Visible | 10 | 1 | 75.0 | [99] |
| Type II $MoS_2$/ZnO | MB | 120 | UV | | 10 | 99.0 | [101] |
| $BiOIO_3$/$MoS_2$ (BM-x) 2D/2D | RhB | 90 | 500 W Xenon lamp | 50 | 10 | 98.7 | [140] |
| ZnO-$MoS_2$ | MR | 60 | Solar | 10 | 10 | 89.0 | [141] |
| $BiFeO_3$/$MoS_2$ | RhB | 200 | Visible | 50 | 10 | 89.0 | [102] |
| ZnO nanorods/$MoS_2$ | RhB | | UV | - | 4 | - | [123] |
| Ultrathin layered $MoS_2$ | MB<br>RhB | 36 | Visible | 100 | 5 | 95.3<br>41.1 | [124] |
| $MoS_2$/$TiO_2$ | MB | 12 | UV-Vis | 50 | 5 | 99.3 | [125] |
| $TiO_2$/$MoS_2$ | MB | 30 | Visible | 10 | 10 | 94.2 | [126] |
| NMS incorporateda-$Fe_2O_3$/ZnO | RhB | 240 | Visible | 40 | | 91.0 | [127] |
| $Ag_3PO_4$/$MoS_2$ | MB | 15 | 60 W CFL | 0.20 g/L | 20 | 97.6 | [128] |
| $MoS_2$/g-$C_3N_4$ | RhB/MO | 60 | Visible | 5 | 50 | 92.0 | [98] |
| PbS/$MoS_2$ | MB | 180 | Visible | 1% | 30 | 83.0 | [129] |
| $MoS_2$QDs@$ZnIn_2S_4$@RGO) | RhB/MB | 30 | 300 W Xenon lamp | 100 | 80 | 98.8<br>98.5 | [130] |
| $SnO_2$-$MoS_2$ | MR/MB | 120 | Visible | 1 | 100 | 58.5<br>94.0 | [131] |
| $MoS_2$-$SrFe_{12}O_{19}$ | RhB | 120 | 300 W Xenon lamp | | 10 | 96.5 | [132] |
| Au-$CoFe_2O_4$/$MoS_2$ | MO | 60 | 300 W iodine tungsten lamp | 70 | 50 | 96.0 | [134] |

## 6. Mechanism of Dye Degradation Using $MoS_2$-Based Systems

$MoS_2$, a semiconductor photocatalyst undergoes electron-hole pair recombination during the degradation reaction. The mechanism of the $MoS_2$-based photocatalyst is described using electron pairs and holes [143]. Based on the different types of electron transfer, the mechanism is classified as traditional and Z-scheme mechanisms. The traditional mechanism of the binary system $MoS_2$/$Cu_2O$ is given in Figure 14a [144], and the ternary system

MoS$_2$/CdS/TiO$_2$ is given in Figure 14c [116]. The Z-scheme mechanism of the binary system MoS$_2$/g-C$_3$N$_4$ [145], and ternary system MoS$_2$/g-C$_3$N$_4$/TiO$_2$ [59] are shown in Figure 14b,d, respectively. In the traditional mechanism, valence band electrons (e$^-$) absorb enough energy to migrate to the conduction band and form holes (h$^+$). Because the conduction band of MoS$_2$ is close to that of TiO$_2$, photoelectrons released in TiO$_2$ are transported to MoS$_2$ more quickly, enabling carrier separation and minimizing the photogenerated electron and hole recombination [126]. In the Z-scheme mechanism [146], the electrons move to CB from VB, and then they will transfer to the VB of the nearest atom. Thus, a zig zag electron transfer is exhibited by these nanocomposites. Mahalakshmi et al. [112] presented an S-scheme mechanism, using MoS$_2$/g-C$_3$N$_4$/TiO$_2$ catalyst (Figure 14e). During the irradiation of light, electrons from VB of g-C$_3$N$_4$, MoS$_2$, and TiO$_2$ are stimulated singly to the CB in this process. In the hole of TiO$_2$'s VB, the excited electrons recombine. The approach helped in the spatial isolation and extraction of photoexcited carriers with larger redox capabilities, and stored electrons of the MoS$_2$/TiO$_2$/g-C$_3$N$_4$ surface are helpful in the decomposition of organic dyes.

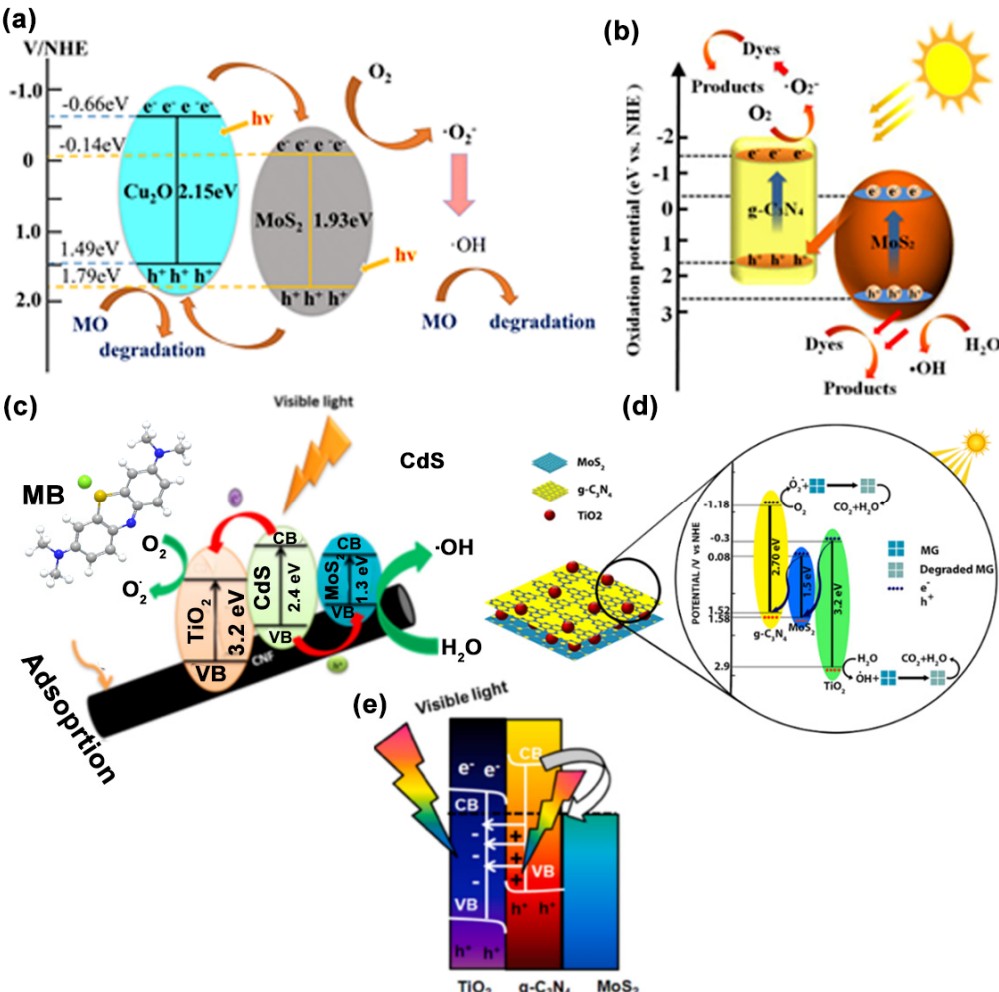

**Figure 14.** The mechanism of the MoS$_2$-based photocatalyst on the dye (**a**) traditional mechanism in binary system MoS$_2$/Cu$_2$O [144], (**b**) Z-scheme mechanism in binary system MoS$_2$/g-C$_3$N$_4$ [145], (**c**) Traditional mechanism in ternary system MoS$_2$/CdS/TiO$_2$ [116], (**d**) Z-scheme in ternary system MoS$_2$/g-C$_3$N$_4$/TiO$_2$ [59], and (**e**) s-scheme in ternary system MoS$_2$/g-C$_3$N$_4$/TiO$_2$ [112]. Reprinted with permission from ref. [59,112,116,144,145]. Copyright 2021-Springer, Copyright 2020-Elsevier, Copyright 2019-Elsevier, Copyright 2020-Springer, Copyright 2017-Elsevier.

As mentioned earlier, the PL spectra have an important role in identifying the particular mechanism of dye degradation. Figure 15a represents the PL spectrum and its traditional mechanism using MoS$_2$@TiO$_2$ [86]. Jia et al. have proposed a Z-scheme mechanism based on the PL spectra for the Au-CoFe$_2$O$_4$/MoS$_2$ nanocomposite [134]. When the spectral intensity is increased after the fabrication, the nanocomposite chooses a Z-scheme mechanism. At low intensities, the composite chooses a traditional mechanism. The PL spectra and the mechanism are given in Figure 15b. A reduction in PL strength indicates that the electron-hole pair recombination has been suppressed.

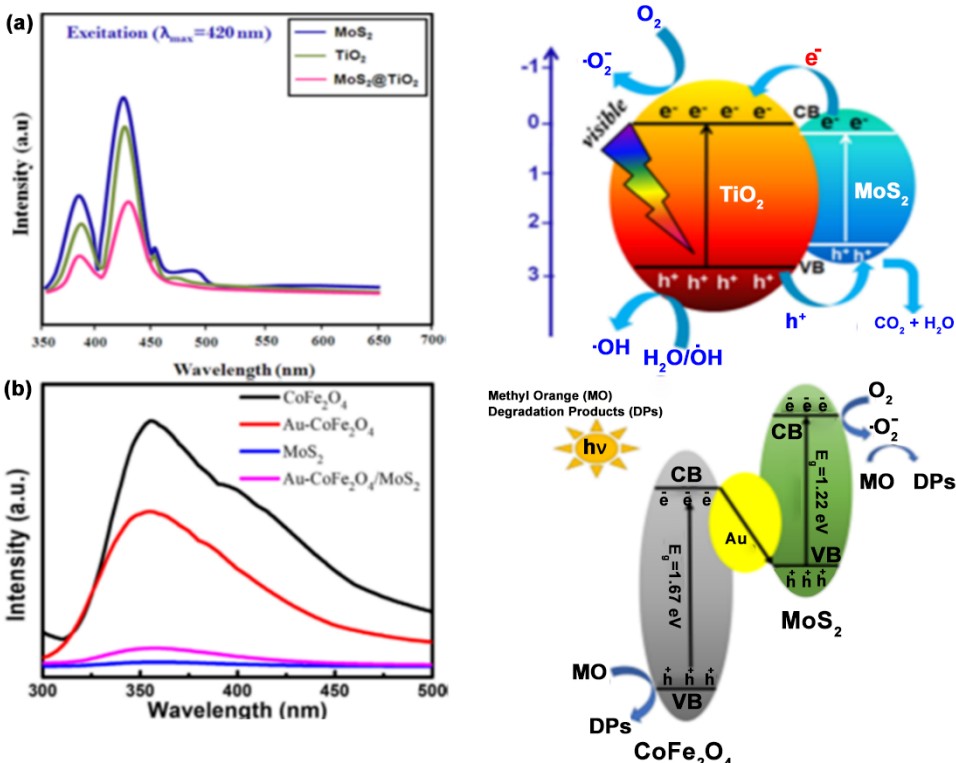

**Figure 15.** PL spectra and their mechanism (**a**) Traditional mechanism of MoS$_2$@ TiO$_2$ [86], (**b**) Z-scheme mechanism of MoS$_2$/CoFe$_2$O$_4$ [134]. Reprinted with permission from ref. [86,134]. Copyright 2020-Elsevier, Copyright 2019-Elsevier.

## 7. Conclusions and Future Perspective

This review has discussed the geometry, polymorphism, different methods of preparation, and characterization, of MoS$_2$-based composites for the photocatalytic degradation of organic dyes. MoS$_2$-based binary and ternary systems have been considered. The degradation ability depends on factors such as morphology and the preparation techniques employed. The chemical properties of the added components help in the formation of a heterojunction with MoS$_2$, leading to an increase in the efficacy of the final catalyst towards the degradation of dyes. Based on the various studies conducted, the photocatalytic activity of MoS$_2$-based nanomaterials has the following characteristics. The force of attraction between the various materials used to fabricate the hybrid catalyst, and the crystallographic plane where the fabrication is taking place on the nanomaterial are significant factors affecting their efficiency. The MoS$_2$-based nanomaterials show great degradation activity towards organic dyes and toxic pollutants. This area of study offers significant opportunities for fabricating excellent photocatalysts to affect the degradation of organic pollutant dyes from wastewater.

**Author Contributions:** J.R.J.U.—Methodology, software, investigation, data curation, writing original draft. M.R.—Methodology, investigation, writing original draft. S.D.K.R.—Investigation, writing, review and editing, supervision, project administration. D.P.—Visualization, validation, writing review and editing. M.K.M.—Resource, investigation, data curation, writing, review and editing. All authors have read and agreed to the published version of the manuscript.

**Funding:** This research received no external funding.

**Institutional Review Board Statement:** Not applicable.

**Informed Consent Statement:** Not applicable.

**Data Availability Statement:** This study did not report any data.

**Acknowledgments:** The authors are grateful to CHRIST (Deemed to be University) for the esteemed support and encouragement.

**Conflicts of Interest:** The authors declare no conflict of interest.

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
