# Peer review of "Structural, Morphological and Optical Properties of MoS2-Based Materials for Photocatalytic Degradation of Organic Dye"

_2673-7256, doi:10.3390/photochem2030042_

Round 1

Reviewer 1 Report

In this review structural information, preparation and characterization methods for MoS2 and composites based on MoS2 have been discussed, pointing the focus on MoS2 degradation activity on pollutants and organic dyes.

The reading of this paper was really a very hard job, because the English text is obscure and needs substantial correction. The presentation and the discussion are rambling, confusing and difficult to read, mainly because of imprecision, incomplete sentences and obscurities. Furthermore there is an incredible number of typos in the tex.

Too many sentences are incomplete and ungrammatical

Some examples:

The most frequently used method for synthesis of flower-like MoS2[56]. Figure 8 represents the preparation steps of MoS2 nanosheets. ( incomplete sentence)

 In 2016, Zhang et al., [100] synthesized a composite from g- C3N4 and exfoliating it with hydrothermally synthesized MoS2 to obtain TiO2/g-C3N4/MoS2 using solvothermal method is used for MO degradation (20 ppm of MO solution)

The structure belongs to the space group. Lattice parameters of the structure is a= b= 3.190 c= 32.319 Å. α=β=90° λ= 120°  Volume is 284.872 Å3.

The differences  in the degradation ability arise from factors such as morphology and the preparation techniques employed. The chemical properties of the other materials used in the fabrication  of the modified MoS2. Based on the….

In summary I cannot recommend publication of the paper. A more precise and refined study is necessary.

Reviewer 2 Report

1. Mention more about transition metal dichalcogenide (TMDCs) in introduction.

2. significant changes in the morphology of MoS2 need to be shown with more experimental results.

Reviewer 3 Report

In the article, the authors presented in detail and compared the preparation methods of catalysts (MoS2 catalysts and composite catalysts containing MoS2) produced by researchers in the recent past. The morphology, structural and optical properties of the catalysts were also presented. In chapter 5, the authors described the photoactivity of the various catalysts in detail. When testing the photoactivity of the produced catalysts, the researchers used different dyes as model compounds in most cases. My questions and comments are related to this part of the article. When presenting and comparing the photoactivity of different catalysts, the authors do not always clearly state the light source used by the researchers. This should be improved so that the reader knows exactly the experimental conditions and the light source used (Although the data I am missing appears in the first table, it should also be included in the text section).

„The band gap saw a red shift after the modification, which helps in the better utilization of visible light/solarlight.” It should be worded more carefully, sunlight also contains UV rays.

„Zhang et al. prepared a ternary system of MoS2 and calculated the band gap as 2.76 eV.” What components does it consist of?

 An important question is whether the investigated catalysts are stable and can be reused. I found a reference to this in only one case.” Pant et al. [105] modified MoS2 using CdS and TiO2, and it was used to examine the breakdown of MB dye on carbon  nanotubes. The dye degraded completely with 15 min of irradiation of visible light. This  system showed a high degradation activity for repeated cycles and can be reused”  Have other researchers not carried out studies in this direction? One of the planned areas of use of the presented catalysts is wastewater treatment. The question arises whether the catalyst does not degrade during use, does it dissolve substances harmful to the organisms living in the water? Do the articles processed by the authors contain information regarding this? If so, it would be good to briefly mention them as well.

 The article contains several minor editing errors, lack of spaces, lack of indexing.

Round 2

Reviewer 1 Report

Although authors have improved the text and the English form, many errors are still present. Below I point out a few.

lines 137, 148, 160 and 163… Lattice parameters of the above structure is are..

line    155 Other geometry: Orthorhombic

line 175 “are listed” instead of  “are described” 

line 207  “150 m” before “m” the symbol micro must be inserted

lines 229-230 3D MoSultrastructure was prepared by Anwer et al. [63] used a controllable …..”     before “used” the pronoun “who” must be inserted

A  more important point concerns the figures.

The resolution of figures in general is poor. In particular the resolution of figs 9, 13 and 14c  has to be improved  before publishing

Reviewer 2 Report

1. Figure 1. Representation of the preparation, morphology, and applications of MoS2 based catalysts make it larger for ease to read.

2. Structure and geometries of MoS2 mention also the electronic properties.

3. Chemical methods mention hydrothermal methods

4. Mention also the MoS2 composite with cerium oxide and its application.

Reviewer 3 Report

I accept corrections and answers, the article can be published in its current form.
